# Fast Two-Sample Testing with Analytic Representations of Probability Measures

**Kacper Chwialkowski**
Gatsby Computational Neuroscience Unit, UCL
kacper.chwialkowski@gmail.com

**Aaditya Ramdas**
Dept. of EECS and Statistics, UC Berkeley
aramdas@cs.berkeley.edu

**Dino Sejdinovic**
Dept of Statistics, University of Oxford
dino.sejdinovic@gmail.com

**Arthur Gretton**
Gatsby Computational Neuroscience Unit, UCL
arthur.gretton@gmail.com

## Abstract

We propose a class of nonparametric two-sample tests with a cost linear in the sample size. Two tests are given, both based on an ensemble of distances between analytic functions representing each of the distributions. The first test uses smoothed empirical characteristic functions to represent the distributions, the second uses distribution embeddings in a reproducing kernel Hilbert space. Analyticity implies that differences in the distributions may be detected almost surely at a finite number of randomly chosen locations/frequencies. The new tests are consistent against a larger class of alternatives than the previous linear-time tests based on the (non-smoothed) empirical characteristic functions, while being much faster than the current state-of-the-art quadratic-time kernel-based or energy distance-based tests. Experiments on artificial benchmarks and on challenging real-world testing problems demonstrate that our tests give a better power/time tradeoff than competing approaches, and in some cases, better outright power than even the most expensive quadratic-time tests. This performance advantage is retained even in high dimensions, and in cases where the difference in distributions is not observable with low order statistics.

## 1 Introduction

Testing whether two random variables are identically distributed without imposing any parametric assumptions on their distributions is important in a variety of scientific applications. These include data integration in bioinformatics [6], benchmarking for steganography [20] and automated model checking [19]. Such problems are addressed in the statistics literature via two-sample tests (also known as homogeneity tests).

Traditional approaches to two-sample testing are based on distances between representations of the distributions, such as density functions, cumulative distribution functions, characteristic functions or mean embeddings in a reproducing kernel Hilbert space (RKHS) [27, 26]. These representations are infinite dimensional objects, which poses challenges when defining a distance between distributions. Examples of such distances include the classical Kolmogorov-Smirnov distance (sup-norm between cumulative distribution functions); the Maximum Mean Discrepancy (MMD) [9], an RKHS norm of the difference between mean embeddings, and the $\mathbb{N}$-distance (also known as energy distance) [34, 31, 4], which is an MMD-based test for a particular family of kernels [25] . Tests may also be based on quantities other than distances, an example being the Kernel Fisher Discriminant (KFD) [12], the estimation of which still requires calculating the RKHS norm of a difference of mean embeddings, with normalization by an inverse covariance operator.

In contrast to consistent two-sample tests, heuristics based on pseudo-distances, such as the difference between characteristic functions evaluated at a single frequency, have been studied in the context of goodness-of-fit tests [13, 14]. It was shown that the power of such tests can be maximized against fully specified alternative hypotheses, where test power is the probability of correctly rejecting the null hypothesis that the distributions are the same. In other words, if the class of distributions being distinguished is known in advance, then the tests can focus only at those particular frequencies where the characteristic functions differ most. This approach was generalized to evaluating the empirical characteristic functions at multiple distinct frequencies by [8], thus improving on tests that need to know the single "best" frequency in advance (the cost remains linear in the sample size, albeit with a larger constant). This approach still fails to solve the consistency problem, however: two distinct characteristic functions can agree on an interval, and if the tested frequencies fall in that interval, the distributions will be indistinguishable.

In Section 2 of the present work, we introduce two novel distances between distributions, which both use a parsimonious representation of the probability measures. The first distance builds on the notion of differences in characteristic functions with the introduction of *smooth characteristic functions*, which can be though of as the analytic analogues of the characteristics functions. A distance between smooth characteristic functions evaluated at a single random frequency is almost surely a distance (Definition 1 formalizes this concept) between these two distributions. In other words, there is no need to calculate the whole infinite dimensional representation - it is almost surely sufficient to evaluate it at a single random frequency (although checking more frequencies will generally result in more powerful tests). The second distance is based on analytic mean embeddings of two distributions in a characteristic RKHS; again, it is sufficient to evaluate the distance between mean embeddings at a single randomly chosen point to obtain almost surely a distance. To our knowledge, this representation is the first mapping of the space of probability measures into a finite dimensional Euclidean space (in the simplest case, the real line) that is almost surely an injection, and as a result almost surely a metrization. This metrization is very appealing from a computational viewpoint, since the statistics based on it have linear time complexity (in the number of samples) and constant memory requirements.

We construct statistical tests in Section 3, based on empirical estimates of differences in the analytic representations of the two distributions. Our tests have a number of theoretical and computational advantages over previous approaches. The test based on differences between analytic mean embeddings is a.s. consistent for all distributions, and the test based on differences between smoothed characteristic functions is a.s. consistent for all distributions with integrable characteristic functions (contrast with [8], which is only consistent under much more onerous conditions, as discussed above). This same weakness was used by [1] in justifying a test that integrates over the *entire* frequency domain (albeit at cost quadratic in the sample size), for which the quadratic-time MMD is a generalization [9]. Compared with such quadratic time tests, our tests can be conducted in linear time – hence, we expect their power/computation tradeoff to be superior.

We provide several experimental benchmarks (Section 4) for our tests. First, we compare test power as a function of computation time for two real-life testing settings: amplitude modulated audio samples, and the Higgs dataset, which are both challenging multivariate testing problems. Our tests give a better power/computation tradeoff than the characteristic function-based tests of [8], the previous sub-quadratic-time MMD tests [11, 32], and the quadratic-time MMD test. In terms of power when unlimited computation time is available, we might expect worse performance for the new tests, in line with findings for linear- and sub-quadratic-time MMD-based tests [15, 9, 11, 32]. Remarkably, such a loss of power is not the rule: for instance, when distinguishing signatures of the Higgs boson from background noise [3] ('Higgs dataset'), we observe that a test based on differences in smoothed empirical characteristic functions outperforms the quadratic-time MMD. This is in contrast to linear- and sub-quadratic-time MMD-based tests, which by construction are less powerful than the quadratic-time MMD. Next, for challenging artificial data (both high-dimensional distributions, and distributions for which the difference is very subtle), our tests again give a better power/computation tradeoff than competing methods.

## 2   Analytic embeddings and distances

In this section we consider mappings from the space of probability measures into a sub-space of real valued analytic functions. We will show that evaluating these maps at $J$ randomly selected

points is almost surely injective for any $J > 0$. Using this result, we obtain a simple (randomized) metrization of the space of probability measures. This metrization is used in the next section to construct linear-time nonparametric two-sample tests.

To motivate our approach, we begin by recalling an integral family of distances between distributions, denoted Maximum Mean Discrepancies (MMD) [9]. The MMD is defined as

$$\text{MMD}(P,Q) = \sup_{f \in B_k} \left[ \int_E f dP - \int_E f dQ \right], \tag{1}$$

where $P$ and $Q$ are probability measures on $E$, and $B_k$ is the unit ball in the RKHS $H_k$ associated with a positive definite kernel $k : E \times E \to \mathbf{R}$. A popular choice of $k$ is the Gaussian kernel $k(x,y) = \exp(-\|x-y\|^2/\gamma^2)$ with bandwidth parameter $\gamma$. It can be shown that the MMD is equal to the RKHS distance between so called mean embeddings,

$$\text{MMD}(P,Q) = \|\mu_P - \mu_Q\|_{H_k}, \tag{2}$$

where $\mu_P$ is an embedding of the probability measure $P$ to $H_k$,

$$\mu_P(t) = \int_E k(x,t) dP(x), \tag{3}$$

and $\| \cdot \|_{H_k}$ denotes the norm in the RKHS $H_k$. When $k$ is translation invariant, i.e., $k(x,y) = \kappa(x-y)$, the squared MMD can be written [27, Corollary 4]

$$\text{MMD}^2(P,Q) = \int_{\mathbf{R}^d} |\varphi_P(t) - \varphi_Q(t)|^2 \, F^{-1}\kappa(t) dt, \tag{4}$$

where $F$ denotes the Fourier transform, $F^{-1}$ is the inverse Fourier transform, and $\varphi_P$, $\varphi_Q$ are the characteristic functions of $P$, $Q$, respectively. From [27, Theorem 9], a kernel $k$ is called *characteristic* when the MMD for $H_k$ satisfies

$$\text{MMD}(P,Q) = 0 \text{ iff } P = Q. \tag{5}$$

Any bounded, continuous, translation-invariant kernel whose inverse Fourier transform is almost everywhere non-zero is characteristic [27]. By representation (2), it is clear that the MMD with a characteristic kernel is a metric.

**Pseudometrics based on characteristic functions.** A practical limitation when using the MMD in testing is that an empirical estimate is expensive to compute, this being the sum of two U-statistics and an empirical average, with cost quadratic in the sample size [9, Lemma 6]. We might instead consider a finite dimensional approximation to the MMD, achieved by estimating the integral (4), with the random variable

$$d_{\varphi,J}^2(P,Q) = \frac{1}{J} \sum_{j=1}^{J} |\varphi_P(T_j) - \varphi_Q(T_j)|^2, \tag{6}$$

where $\{T_j\}_{j=1}^{J}$ are sampled independently from the distribution with a density function $F^{-1}\kappa$. This type of approximation is applied to various kernel algorithms under the name of *random Fourier features* [21, 17]. In the statistical testing literature, the quantity $d_{\varphi,J}(P,Q)$ predates the MMD by a considerable time, and was studied in [13, 14, 8], and more recently revisited in [33]. Our first proposition is that $d_{\varphi,J}^2(P,Q)$ can be a poor choice of distance between probability measures, as it fails to distinguish a large class of measures. The following result is proved in the Appendix.

**Proposition 1.** *Let $J \in \mathbb{N}$ and let $\{T_j\}_{j=1}^{J}$ be a sequence of real valued i.i.d. random variables with a distribution which is absolutely continuous with respect to the Lebesgue measure. For any $0 < \epsilon < 1$, there exists an uncountable set $\mathcal{A}$ of mutually distinct probability measures (on the real line) such that for any $P, Q \in \mathcal{A}$, $\mathbb{P}\left(d_{\varphi,J}^2(P,Q) = 0\right) \geq 1 - \epsilon$.*

We are therefore motivated to find distances of the form (6) that can distinguish larger classes of distributions, yet remain efficient to compute. These distances are characterized as follows:

**Definition 1** (Random Metric). *A random process $d$ with values in $\mathbf{R}$, indexed with pairs from the set of probability measures $\mathcal{M}$, i.e., $d = \{d(P,Q) : P, Q \in \mathcal{M}\}$, is said to be a random metric if it satisfies all the conditions for a metric with qualification 'almost surely'. Formally, for all $P, Q, R \in \mathcal{M}$, random variables $d(P,Q), d(P,R), d(R,Q)$ must satisfy*

1. $d(P, Q) \geq 0$ *a.s.*

2. *if* $P = Q$, *then* $d(P, Q) = 0$ *a.s, if* $P \neq Q$ *then* $d(P, Q) \neq 0$ *a.s.*

3. $d(P, Q) = d(Q, P)$ *a.s.*

4. $d(P, Q) \leq d(P, R) + d(R, Q)$ *a.s.* [1]

From the statistical testing point of view, the coincidence axiom of a metric $d$, $d(P, Q) = 0$ if and only if $P = Q$, is key, as it ensures consistency against all alternatives. The quantity $d_{\varphi, J}(P, Q)$ in (6) violates the coincidence axiom, so it is only a random pseudometric (other axioms are trivially satisfied). We remedy this problem by replacing the characteristic functions by smooth characteristic functions:

**Definition 2.** *A smooth characteristic function $\phi_P(t)$ of a measure $P$ is a characteristic function of $P$ convolved with an analytic smoothing kernel $l$, i.e.*

$$\phi_P(t) = \int_{\mathbf{R}^d} \varphi_P(w) l(t - w) dw, \qquad t \in \mathbf{R}^d. \tag{7}$$

Proposition 3 shows that smooth characteristic function can be estimated in a linear time. The analogue of $d_{\varphi, J}(P, Q)$ for smooth characteristic functions is simply

$$d_{\phi, J}^2(P, Q) = \frac{1}{J} \sum_{j=1}^{J} |\phi_P(T_j) - \phi_Q(T_j)|^2, \tag{8}$$

where $\{T_j\}_{j=1}^{J}$ are sampled independently from the absolutely continuous distribution (returning to our earlier example, this might be $F^{-1}\kappa(t)$ if we believe this to be an informative choice). The following theorem, proved in the Appendix, demonstrates that the smoothing greatly increases the class of distributions we can distinguish.

**Theorem 1.** *Let $l$ be an analytic, integrable kernel with an inverse Fourier transform that is non-zero almost everywhere. Then, for any $J > 0$, $d_{\phi, J}$ is a random metric on the space of probability measures with integrable characteristic functions, and $\phi_P$ is an analytic function.*

This result is primarily a consequence of analyticity of smooth characteristic functions and the fact that analytic functions are 'well behaved'. There is an additional, practical advantage to smoothing: when the variability in the difference of the characteristic functions is high, and these differences are local, smoothing distributes the difference in CFs more broadly in the frequency domain (a simple illustration is in Fig. A.1, Appendix), making it easier to find by measurement at a small number of randomly chosen points. This accounts for the observed improvements in test power in Section 4, over differences in unsmoothed CFs.

**Metrics based on mean embeddings.** The key step which leads us to the construction of a random metric $d_{\phi, J}$ is the convolution of the original characteristic functions with an analytic smoothing kernel. This idea need not be restricted to the representations of probability measures in the frequency domain. We may instead directly convolve the probability measure with a positive definite kernel $k$ (that need not be translation invariant), yielding its mean embedding into the associated RKHS,

$$\mu_P(t) = \int_E k(x, t) dP(x). \tag{9}$$

We say that a positive definite kernel $k : \mathbf{R}^D \times \mathbf{R}^D \to \mathbf{R}$ is analytic on its domain if for all $x \in \mathbf{R}^D$, the feature map $k(x, \cdot)$ is an analytic function on $\mathbf{R}^D$. By using embeddings with *characteristic and analytic* kernels, we obtain particularly useful representations of distributions. As for the smoothed CF case, we define

$$d_{\mu, J}^2(P, Q) = \frac{1}{J} \sum_{j=1}^{J} (\mu_P(T_j) - \mu_Q(T_j))^2. \tag{10}$$

The following theorem ensures that $d_{\mu, J}(P, Q)$ is also a random metric.

**Theorem 2.** *Let $k$ be an analytic, integrable and characteristic kernel. Then for any $J > 0$, $d_{\mu,J}$ is a random metric on the space of probability measures (and $\mu_P$ is an analytic function).*

Note that this result is stronger than the one presented in Theorem 1, since it is not restricted to the class of probability measures with integrable characteristic functions. Indeed, the assumption that the characteristic function is integrable implies the existence and boundedness of a density. Recalling the representation of MMD in (2), we have proved that it is almost always sufficient to measure difference between $\mu_P$ and $\mu_Q$ at a finite number of points, provided our kernel is characteristic and analytic. In the next section, we will see that metrization of the space of probability measures using random metrics $d_{\mu,J}$, $d_{\phi,J}$ is very appealing from the computational point of view. It turns out that the statistical tests that arise from these metrics have linear time complexity (in the number of samples) and constant memory requirements.

## 3   Hypothesis Tests Based on Distances Between Analytic Functions

In this section, we provide two linear-time two-sample tests: first, a test based on analytic mean embeddings, and next a test based on smooth characteristic functions. We further describe the relation with competing alternatives. Proofs of all propositions are in Appendix B.

**Difference in analytic functions**  In the previous section we described the random metric based on a difference in analytic mean embeddings, $d_{\mu,J}^2(P,Q) = \frac{1}{J} \sum_{j=1}^{J} (\mu_P(T_j) - \mu_Q(T_j))^2$. If we replace $\mu_P$ with the empirical mean embedding $\hat{\mu}_P = \frac{1}{n} \sum_{i=1}^{n} k(X_i, \cdot)$ it can be shown that for any sequence of unique $\{t_j\}_{j=1}^{J}$, under the null hypothesis, as $n \to \infty$,

$$\sqrt{n} \sum_{j=1}^{J} (\hat{\mu}_P(t_j) - \hat{\mu}_Q(t_j))^2 \tag{11}$$

converges in distribution to a sum of correlated chi-squared variables. Even for fixed $\{t_j\}_{j=1}^{J}$, it is very computationally costly to obtain quantiles of this distribution, since this requires a bootstrap or permutation procedure. We will follow a different approach based on Hotelling's $T^2$-statistic [16]. The Hotelling's $T^2$-squared statistic of a normally distributed, zero mean, Gaussian vector $W = (W_1, \cdots, W_J)$, with a covariance matrix $\Sigma$, is $T^2 = W\Sigma^{-1}W$. The compelling property of the statistic is that it is distributed as a $\chi^2$-random variable with $J$ degrees of freedom. To see a link between $T^2$ and equation (11), consider a random variable $\sum_{i=j}^{J} W_j^2$: this is also distributed as a sum of correlated chi-squared variables. In our case $W$ is replaced with a difference of normalized empirical mean embeddings, and $\Sigma$ is replaced with the empirical covariance of the difference of mean embeddings. Formally, let $Z_i$ denote the vector of differences between kernels at tests points $T_j$,

$$Z_i = (k(X_i, T_1) - k(Y_i, T_1), \cdots, k(X_i, T_J) - k(Y_i, T_J)) \in \mathbf{R}^J. \tag{12}$$

We define the vector of mean empirical differences $W_n = \frac{1}{n} \sum_{i=1}^{n} Z_i$, and its covariance matrix $\Sigma_n = \frac{1}{n} \sum_i (Z_i - W_n)(Z_i - W_n)^T$. The test statistic is

$$S_n = nW_n\Sigma_n^{-1}W_n. \tag{13}$$

The computation of $S_n$ requires inversion of a $J \times J$ matrix $\Sigma_n$, but this is fast and numerically stable: $J$ will typically be small, and is less than 10 in our experiments. The next proposition demonstrates the use of $S_n$ as a two-sample test statistic.

**Proposition 2** (Asymptotic behavior of $S_n$)**.** *Let $d_{\mu,J}^2(P,Q) = 0$ a.s. and let $\{X_i\}_{i=1}^{n}$ and $\{Y_i\}_{i=1}^{n}$ be i.i.d. samples from $P$ and $Q$ respectively. If $\Sigma_n^{-1}$ exists for $n$ large enough, then the statistic $S_n$ is a.s. asymptotically distributed as a $\chi^2$-random variable with $J$ degrees of freedom (as $n \to \infty$ with $d$ fixed). If $d_{\mu,J}^2(P,Q) > 0$ a.s., then a.s. for any fixed $r$, $\mathbb{P}(S_n > r) \to 1$ as $n \to \infty$.*

We now apply the above proposition to obtain a statistical test.

**Test 1** (Analytic mean embedding )**.** *Calculate $S_n$. Choose a threshold $r_\alpha$ corresponding to the $1 - \alpha$ quantile of a $\chi^2$ distribution with $J$ degrees of freedom, and reject the null hypothesis whenever $S_n$ is larger than $r_\alpha$.*

There are a number of valid sampling schemes for the test points $\{T_j\}_{j=1}^J$ to evaluate the differences in mean embeddings: see Section 4 for a discussion.

**Difference in smooth characteristic functions** From the convolution definition of a smooth characteristic function (7) it is not immediately obvious how to calculate its estimator in linear time. In the next proposition, however, we show that a smooth characteristic function is an expected value of some function (with respect to the given measure), which can be estimated in a linear time.

**Proposition 3.** *Let $k$ be an integrable translation-invariant kernel and $f$ its inverse Fourier transform. Then the smooth characteristic function of $P$ can be written as $\phi_P(t) = \int_{\mathbf{R}^d} e^{it^\top x} f(x) dP(x)$.*

It is now clear that a test based on the smooth characteristic functions is similar to the test based on mean embeddings. The main difference is in the definition of the vector of differences $Z_i$:

$$Z_i = (f(X_i)\sin(X_iT_1) - f(Y_i)\sin(Y_iT_1), f(X_i)\cos(X_iT_1) - f(Y_i)\cos(Y_iT_1), \cdots) \in \mathbf{R}^{2J} \quad (14)$$

The imaginary and real part of the $e^{\sqrt{-1}T_j^\top X_i} f(X_i) - e^{\sqrt{-1}T_j^\top Y_i} f(Y_i)$ are stacked together, in order to ensure that $W_n$, $\Sigma_n$ and $S_n$ as all real-valued quantities.

**Proposition 4.** *Let $d^2_{\phi,J}(P,Q) = 0$ and let $\{X_i\}_{i=1}^n$ and $\{Y_i\}_{i=1}^n$ be i.i.d. samples from $P$ and $Q$ respectively. Then the statistic $S_n$ is almost surely asymptotically distributed as a $\chi^2$-random variable with $2J$ degrees of freedom (as $n \to \infty$ with $J$ fixed). If $d^2_{\phi,J}(P,Q) > 0$, then almost surely for any fixed $r$, $P(S_n > r) \to 1$ as $n \to \infty$.*

**Other tests**. The test [8] based on empirical characteristic functions was constructed originally for one test point and then generalized to many points - it is quite similar to our second test, but does not perform smoothing (it is also based on a $T^2$-Hotelling statistic). The block MMD [32] is a sub-quadratic test, which can be trivially linearized by fixing the block size, as presented in the Appendix. Finally, another alternative is the MMD, an inherently quadratic time test. We scale MMD to linear time by sub-sampling our data set, and choosing only $\sqrt{n}$ points, so that the MMD complexity becomes $O(n)$. Note, however, that the true complexity of MMD involves a permutation calculation of the null distribution at cost $O(b_n n)$, where the number of permutations $b_n$ grows with $n$. See Appendix C for a detailed description of alternative tests.

# 4  Experiments

In this section we compare two-sample tests on both artificial benchmark data and on real-world data. We denote the smooth characteristic function test as 'Smooth CF', and the test based on the analytic mean embeddings as 'Mean Embedding'. We compare against several alternative testing approaches: block MMD ('Block MMD'), a characteristic functions based test ('CF'), a sub-sampling MMD test ('MMD($\sqrt{n}$)'), and the quadratic-time MMD test ('MMD(n)').
**Experimental setup.** For all the experiments, $D$ is the dimensionality of samples in a dataset, $n$ is a number of samples in the dataset (sample size) and $J$ is number of test frequencies. Parameter selection is required for all the tests. The table summarizes the main choices of the parameters made for the experiments. The first parameter is the test function, used to calculate the particular statistic. The scalar $\gamma$ represents the length-scale of the observed data. Notice that for the kernel tests we recover the standard parameterization $\exp(-\|\frac{x}{\gamma} - \frac{y}{\gamma}\|^2) = \exp(-\frac{\|x-y\|^2}{\gamma^2})$. The original CF test was proposed without any parameters, hence we added $\gamma$ to ensure a fair comparison - for this test varying $\gamma$ is equivalent to adjusting the variance of the distribution of frequencies $T_j$. For all tests, the value of the scaling parameter $\gamma$ was chosen so as to minimize a p-value estimate on a held-out training set: details are described in Appendix D. We chose not to optimize the sampling scheme for the Mean Embedding and Smooth CF tests, since this would give them an unfair advantage over the Block MMD, MMD($\sqrt{n}$) and CF tests. The block size in the Block MMD test and the number of test frequencies in the Mean Embedding, Smooth CF, and CF tests, were always set to the same value (not greater than 10) to maintain exactly the same time complexity. Note that we did *not* use the popular median heuristic for kernel bandwidth choice (MMD and B-test), since it gives poor results for the Blobs and AM Audio datasets [11]. We do not run MMD(n) test for 'Simulation 1' or 'Amplitude Modulated Music', since the sample size is 10000, and too large for a quadratic-time test with permutation sampling for the test critical value.

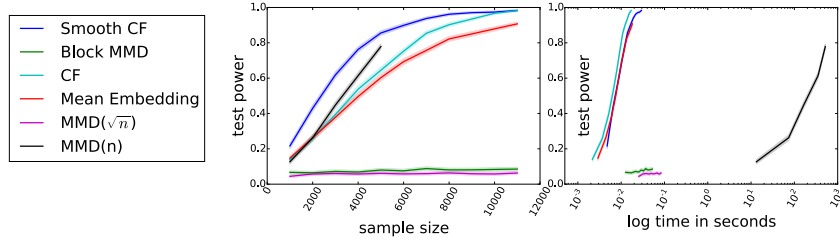

Figure 1: Higgs dataset. **Left:** Test power vs. sample size. **Right:** Test power vs. execution time.

It is important to verify that Type I error is indeed at the design level, set at $\alpha = 0.05$ in this paper. This is verified in the Appendix, Figure A.2. Also shown in the plots is the $95\%$ percent confidence intervals for the results, as averaged over 4000 runs.

| Test | Test Function | Sampling scheme | Other parameters |
|---|---|---|---|
| Mean Embedding | $\exp(-\|\gamma^{-1}(x-t)\|^2)$ | $T_j \sim N(0_D, I_D)$ | $J$ - no. of test frequencies |
| Smooth CF | $\exp(it^\top \gamma^{-1}x - \|\gamma^{-1}x - t\|^2)$ | $T_j \sim N(0_D, I_D)$ | $J$ - no. of test frequencies |
| MMD(n),MMD($\sqrt{n}$) | $\exp(-\|\gamma^{-1}(x-t)\|^2)$ | not applicable | $b$ -bootstraps |
| Block MMD | $\exp(-\|\gamma^{-1}(x-t)\|^2)$ | not applicable | $B$-block size |
| CF | $\exp(it^\top \gamma^{-1}x)$ | $T_j \sim N(0_D, I_D)$ | $J$ - no. of test frequencies |

**Real Data 1: Higgs dataset,** $D = 4$, n varies, $J = 10$. The first experiment we consider is on the UCI Higgs dataset [18] described in [3] - the task is to distinguish signatures of processes that produce Higgs bosons from background processes that do not. We consider a two-sample test on certain extremely low-level features in the dataset - kinematic properties measured by the particle detectors, i.e., the joint distributions of the azimuthal angular momenta $\varphi$ for four particle jets. We denote by $P$ the jet $\varphi$-momenta distribution of the background process (no Higgs bosons), and by $Q$ the corresponding distribution for the process that produces Higgs bosons (both are distributions on $\mathbf{R}^4$). As discussed in [3, Fig. 2], $\varphi$-momenta, unlike transverse momenta $p_T$, carry very little discriminating information for recognizing whether Higgs bosons were produced. Therefore, we would like to test the null hypothesis that the distributions of angular momenta $P$ (no Higgs boson observed) and $Q$ (Higgs boson observed) might yet be rejected. The results for different algorithms are presented in the Figure 1. We observe that the joint distribution of the angular momenta is in fact discriminative. Sample size varies from 1000 to 12000. The Smooth CF test has significantly higher power than the other tests, including the quadratic-time MMD, which we could only run on up to 5100 samples due to computational limitations. The leading performance of the Smooth CF test is especially remarkable given it is several orders of magnitude faster than the quadratic-time MMD(n), even though we used the fastest quadratic-time MMD implementation, where the asymptotic distribution is approximated by a Gamma density .

**Real Data 2: Amplitude Modulated Music,** $D = 1000$, $n = 10000$, $J = 10$. Amplitude modulation is the earliest technique used to transmit voice over the radio. In the following experiment observations were one thousand dimensional samples of carrier signals that were modulated with two different input audio signals from the same album, song $P$ and song $Q$ (further details of these data are described in [11, Section 5]). To increase the difficulty of the testing problem, independent Gaussian noise of increasing variance (in the range 1 to 4.0) was added to the signals. The results are presented in the Figure 2. Compared to the other tests, the Mean Embedding and Smooth CF tests are more robust to the moderate noise contamination.

**Simulation 1: High Dimensions,** $D$ varies, $n = 10000$, $J = 3$. It has recently been shown, in theory and in practice, that the two-sample problem gets more difficult for an increasing number of dimensions increases on which the distributions do not differ [22, 23]. In the following experiment, we study the power of the two-sample tests as a function of dimension of the samples. We run two-sample tests on two datasets of Gaussian random vectors which differ *only* in the first dimension,

$$\text{Dataset I:} \quad P = N(0_D, I_D) \qquad vs. \qquad Q = N((1, 0, \cdots, 0), I_D)$$
$$\text{Dataset II:} \quad P = N(0_D, I_D) \qquad vs. \qquad Q = N(0_D, \text{diag}((2, 1, \cdots, 1))),$$

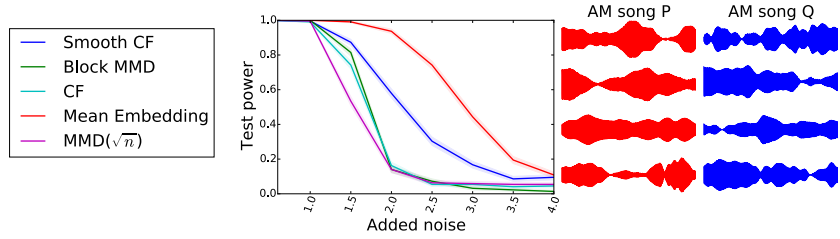

Figure 2: Music Dataset.**Left:** Test power vs. added noise. **Right:** four samples from $P$ and $Q$.

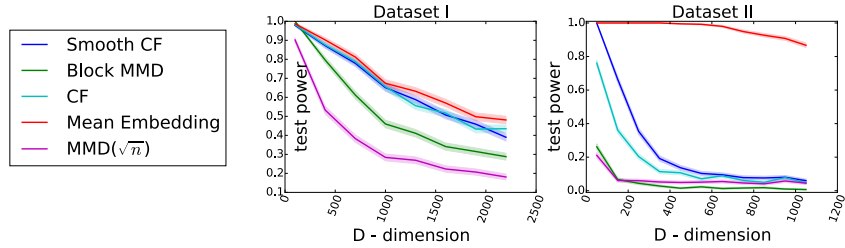

Figure 3: Power vs. redundant dimensions comparison for tests on high dimensional data.

where $0_d$ is a $D$-dimensional vector of zeros, $I_D$ is a $D$-dimensional identity matrix, and diag$(v)$ is a diagonal matrix with $v$ on the diagonal. The number of dimensions (D) varies from 50 to 2500 (Dataset I) and from 50 to 1200 (Dataset II). The power of the different two-sample tests is presented in Figure 3. The Mean Embedding test yields best performance for both datasets, where the advantage is especially large for differences in variance.

**Simulation 2: Blobs,** $D = 2$, *n varies, $J = 5$.* The Blobs dataset is a grid of two dimensional Gaussian distributions (see Figure 4), which is known to be a challenging two-sample testing task. The difficulty arises from the fact that the difference in distributions is encoded at a much smaller lengthscale than the overall data. In this experiment both $P$ and $Q$ are four by four grids of Gaussians, where $P$ has unit covariance matrix in each mixture component, while each component of $Q$ has direction of the largest variance rotated by $\pi/4$ and amplified to 4. It was demonstrated by [11] that a good choice of kernel is crucial for this task. Figure 4 presents the results of two-sample tests on the Blobs dataset. The number of samples varies from 50 to 14000 ( MMD(n) reached test power one with $n = 1400$). We found that the MMD(n) test has the best power as function of the sample size, but the worst power/computation tradeoff. By contrast, random distance based tests have the best power/computation tradeoff.

**Acknowledgment.** We would like thank Bharath Sriperumbudur and Wittawat Jitkrittum for insightful comments.

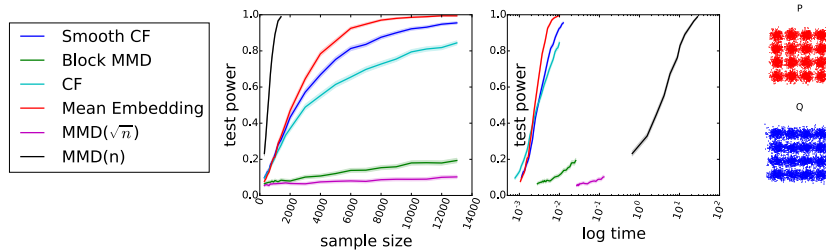

Figure 4: Blobs Dataset. **Left:** test power vs. sample size. **Center:** test power vs. execution time. **Right:** illustration of the blob dataset.

## Footnotes

[1] Note that this does not imply that realizations of $d$ are distances on $\mathcal{M}$, but it does imply that they are almost surely distances for all arbitrary finite subsets of $\mathcal{M}$.

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
