[Supplementary Material]

## A    Figures

Figure A.1: **Smooth vs non-smooth. Left**: pseudo-distance $d_{\varphi,1}(P,Q)$ which uses a single frequency $t \in \mathbf{R}^2$ as a function of this frequency; **Middle**: $d_{\phi,1}(P,Q)$ depicted in the same way; **Right**: $d_{\mu,1}(P,Q)$ which uses a single location $t \in \mathbf{R}^2$ as a function of this location. The measures $P,Q$ used are illustrated in Figure 4 - these are grids of Gaussian distributions discussed in detail in Section 4.

Figure A.2: Type I error of the blobs dataset **(left)** and the dimensions dataset **(right)**. The dashed line is the 99% Wald interval $\alpha \pm 2.57\sqrt{\alpha(1-\alpha)/4000}$ (4000 is number of repetitions) around the design test size of $\alpha = 0.05$.

## B    Proofs

**Proof of Proposition 1**

*Proof.* We write as $P(T)$ the probability measure from which $\{T_j\}_{j=1}^J$ are drawn i.i.d. For some $I = I(\epsilon)$, we specify that there exists an interval $[-I, I]$ where $P(T)$ has mass $1 - (1-\epsilon)^{\frac{1}{J}}$. Define $f_w(t) = 1 - w|t|$ for $w > \frac{1}{I}$ and zero elsewhere. By Polya's theorem, $\mathcal{A} = \{f_w\}_{w > \frac{1}{I}}$ is an uncountable family of characteristic functions that are the same on the complement of $[-I, I]$, which has measure $(1-\epsilon)^{\frac{1}{J}}$. For $w_1 > w_2 > \frac{1}{I}$, $f_{w_1} \neq f_{w_2}$ in some neighborhood of $1/w_1$, hence the measures associated with those characteristic functions are different. The probability that all $T_i$ sit in the complement of interval $[-I, I]$ is $\left((1-\epsilon)^{\frac{1}{J}}\right)^J = (1-\epsilon)$ and such an event implies that $S_{\varphi,J}^2 = 0$.

$\square$

**Proof of Theorem 2**

First we give a proposition that characterizes limits of analytic functions.

**Proposition 5** ( [7, Proposition 3] )**.** *If $\{f_n\}$ is a sequence of real valued, uniformly bounded $(\exists_M \forall_n \|f_n\|_\infty \leq M)$, analytic functions on $\mathbf{R}^d$ converging pointwise to $f$, then $f$ is analytic.*

Now we characterize the RKHS of an analytic kernel. Similar results were proved for specific classes of kernels in [30, Theorem 1], [29, Corollary 3.5].

**Lemma 1.** *If $k$ is a bounded, analytic kernel on $\mathbf{R}^d \times \mathbf{R}^d$, then all functions in the RKHS $\mathcal{H}_k$ associated with this kernel are analytic.*

*Proof.* Since $\mathbf{R}^d$ is separable then by [28, Lemma 4.33] Hilbert Space $\mathcal{H}_k$ is separable. By Moore-Aronszajn Theorem [5] there exist a set $H_0$ of linear combinations of functions $k(\cdot, x), x \in \mathbf{R}^d$, which is dense in $\mathcal{H}_k$ and $\mathcal{H}_k$ is a set of functions which are pointwise limits of Cauchy sequences in $H_0$. For each $f \in \mathcal{H}_k$ let $\{f_n\} \in \mathcal{H}_0$ be a sequence of functions converging in the Hilbert Space norm to $f$. Since $\{f_n\}$ is convergent there exists $N$ such that $\forall n > N \ \|f_n - f\| \leq 1$. For all $n$ there exist a uniform bound on $f_n$ norm

$$\|f_n\| = \|f_n - f + f\| \leq \|f_n - f\| + \|f\| \leq \max(1, \max_{1 \leq i \leq N} \|f_N\|) + \|f\|. \tag{15}$$

Since $k$ is bounded, by the [28, Lemma 4.33], there exists $C$ such that for any $f \in \mathcal{H}_k$, $\|f\|_\infty \leq C\|f\|$. Therefore for all $n$

$$\|f_n\|_\infty \leq C \max(1, \max_{1 \leq i \leq N} \|f_N\|) + C\|f\|. \tag{16}$$

Finally, using Proposition 5 we conclude that $f$ is analytic.

$\square$

Next, we show that analytic functions are 'well behaved'.

**Lemma 2.** *Let $\mu$ be absolutely continuous measure on $\mathbf{R}^d$ (wrt. the Lebesgue measure). Non-zero, analytic function $f$ can be zero at most at the set of measure 0, with respect to the measure $\mu$.*

*Proof.* If $f$ is zero at the set with a limit point then it is zero everywhere. Therefore $f$ can be zero at most at a set $A$ without a limit point, which by definition is a discrete set (distance between any two points in $A$ is greater then some $\epsilon > 0$). Discrete sets have zero Lebesgue measure (as a countable union of points with zero measure). Since $P$ is absolutely continuous then $\mu(A)$ is zero as well. $\square$

Next, we show how to construct random distances.

**Lemma 3.** *Let $\Lambda$ be an injective mapping from the space of the probability measures into a space of analytic functions on $\mathbf{R}^d$. Define*

$$d_{\Lambda,J}^2(P,Q) = \sum_{j=1}^J \left| [\Lambda P](T_j) - [\Lambda Q](T_j) \right|^2$$

*where $\{T_j\}_{j=1}^J$ are real valued i.i.d. random variables from a distribution which is absolutely continuous with respect to the Lebesgue measure. Then, $d_{\Lambda,J}^2(P,Q)$ is a random metric.*

*Proof.* Let $\Lambda P$ and $\Lambda Q$ be images of measures $P$ and $Q$ respectively. We want to apply Lemma 2 to the analytic function $f = \Lambda P - \Lambda Q$, with the measure $\mu = \mu_{T_i}$, to see that if $P \neq Q$ then $f \neq 0$ a.s. To do so, we need to show that $P \neq Q$ implies that $f$ is non-zero. Since mapping to $\Lambda$ is injective, there must exists at least one point $o$ where $f$ is non-zero. By continuity of $f$, there exists a ball around $o$ in which $f$ is non-zero.

We have shown that $P \neq Q$ implies $f \neq 0$ a.s. which in turn implies that $d_{\Lambda,J}(P,Q) > 0$ a.s. If $P = Q$ then $f = 0$ and $d_{\Lambda,J}(P,Q) = 0$.

By the construction $d_{\Lambda,J}(P,Q) = d_{\Lambda,J}(Q,P)$ and for any measure $U$, $d_{\Lambda,J}(P,Q) \leq d_{\Lambda,J}(P,U) + d_{\Lambda,J}(U,Q)$ a.s. since the triangle inequality holds for any vectors in $\mathbf{R}^J$. $\square$

We are ready to prove Theorem 2.

*Proof of Theorem 2.* Since $k$ is characteristic the mapping $\Lambda : P \to \mu_P$ is injective. Since $k$ is a bounded, analytic kernel on $\mathbf{R}^d \times \mathbf{R}^d$, the Lemma 1 guarantees that $\mu_P$ is analytic, hence the image of $\Lambda$ is a subset of analytic functions. Therefore, we can use Lemma 3 to see that $d_{\Lambda,J}(P,Q)^2 = d_{\mu,J}(P,Q)^2$ is a random metric. $\square$

**Proof of Theorem 1**

We first show that smooth characteristic functions are unique to distributions.

**Lemma 4.** *If $l$ is an analytic, integrable, translation invariant kernel with an inverse Fourier transform non-zero almost everywhere and $P$ has integrable characteristic function, then the mapping*

$$\Lambda : P \rightarrow \phi_P$$

*is injective and $\phi_P$ is element of the RKHS $\mathcal{H}_l$ associated with $l$.*

*Proof.* For the integrable characteristic function $\varphi$ we define a functional $L : \mathcal{H}_l \rightarrow R$ given by formula

$$Lf = \int_{\mathbf{R}^d} \varphi(x) f(x) dx \tag{17}$$

Since $L(f+g) = L(f) + L(g)$, $L$ is linear. We check that $L$ is bounded; let $B = \{f \in \mathcal{H}_l : \parallel f \parallel \leq 1\}$ be a unit ball in the Hilbert Space.

$$\sup_{f \in B} |Lf| \leq \sup_{f \in B} \int_{\mathbf{R}^d} \varphi(x) f(x) dx \leq \sup_{f \in B} \int_{\mathbf{R}^d} \varphi(x) \|f\| l(x,x) dx = \int_{\mathbf{R}^d} \varphi(x) l(x,x) dx \leq \infty \tag{18}$$

By Riesz representation Theorem there exist $\phi \in H$ such that $\langle \phi, f \rangle = \int_{\mathbf{R}^d} \varphi(x) f(x) dx$. By reproducing property $\phi$ is given by equation $\phi(x) = \langle \phi, l(t,) \rangle = \int_{\mathbf{R}^d} l(x,t) \varphi(x) dx$. With each probability measure $P$ with an integrable characteristic function $\varphi_P$ we associate the smooth characteristic function with

$$P \rightarrow \phi_P(x) = \int_{\mathbf{R}^d} l(x,t) \varphi_P(x) dx \tag{19}$$

We will prove that $P \rightarrow \phi_P$ is injective. We will show that, $\forall_x \phi_Q(x) = \phi_P(x)$ implies $P = Q$.

$$\phi_Q = \phi_P \Rightarrow \int_{\mathbf{R}^d} l(x-t) \varphi_P(x) dx = \int_{\mathbf{R}^d} l(x-t) \varphi_Q(x) dx. \tag{20}$$

We apply inverse Fourier transform to this convolution and get

$$g(x) f_X(x) = f_Y(x) g(x) \tag{21}$$

Where $g = T^{-1} l$, $f_Y = T^{-1} \varphi_Q$ and $f_X = T^{-1} \varphi_P$. Since inverse Fourier transform is injective on the space of the integrable characteristic functions, and all $l, \varphi_P, \varphi_Q$ are integrable CFs, then application of the inverse Fourier transform does not enlarge the null space of Eq. (20). Since $g(x)$ is non-zero almost everywhere, and $f_X$, $f_Y$ are densities, $f_X(x) = f_Y(x)$ almost everywhere, implying that the mapping $P \rightarrow \phi_P$ is injective.

$\square$

Next, we show that smooth characteristic function is analytic.

**Lemma 5.** *If $l$ is an analytic, integrable kernel with an inverse Fourier transform non-zero almost everywhere and $P$ has an integrable characteristic function then the smooth characteristic function $\phi_P$ is analytic.*

*Proof.* By lemma 3, all functions in the RKHS associated with $l$ are analytic, and by Lemma 4 $\phi_P$ is an element of this RKHS. $\square$

We are ready to prove Theorem 1.

*Proof of Theorem 1.* Since $l$ is an analytic, integrable kernel with an inverse Fourier transform non-zero almost everywhere then by the Lemma 4 the mapping $\Lambda : P \rightarrow \phi_P$ is injective and $\Lambda(P)$ is an element of the RKHS associated with $l$. Lemma 5 shows that $\mu_P$ is analytic. Therefore we can use Lemma 3 to see that $d_{\Lambda,J}(P,Q)^2 = d_{\phi,J}(P,Q)^2$ is a random metric. $\square$

**Proof of Lemma 3**

*Proof.* By Fubini's theorem we get

$$\phi_P(t) = \int_{\mathbf{R}^d} \varphi_P(t-w) f(w) dw$$

$$= \int_{\mathbf{R}^d} \left( \int_{\mathbf{R}^d} e^{i(t-w)^\top x} dP(x) \right) f(w) dw$$

$$= \int_{\mathbf{R}^d} e^{it^\top x} \left( \int_{\mathbf{R}^d} e^{-iw^\top x} f(w) dw \right) dP(x)$$

$$= \mathbb{E}[e^{it^\top X} F f(X)].$$

Use of Fubini's theorem is justified, since the iterated integral is finite [24][Theorem 8.8 b] i.e.

$$\int_{\mathbf{R}^d} \int_{\mathbf{R}^d} |e^{i(t-w)^\top x} f(w)| dP(x) dw$$

$$= \int_{\mathbf{R}^d} |f(w)| \int_{\mathbf{R}^d} 1 dP(x) dw < \infty.$$

□

**Proof of Proposition 2**

*Proof.* The probability space of random variables $\{T_j\}_{1 \le j \le J}$ and $\{X_i\}_{1 \le i \le n}$ is a product space i.e sequence of $T_j$'s is defined on the space $(\Omega_1, \mathcal{F}_1, P_1)$ and the sequence of $X_i$'s is defined on the space $(\Omega_2, \mathcal{F}_2, P_2)$. We will show that for almost all $\omega \in \Omega_1$, $S_n$ converges to $\chi^2$ distribution with $J$ degrees of freedom. We define

$$Z_i^\omega = (k(X_i, T_1(\omega)) - k(Y_i, T_1(\omega)), \cdots, k(X_i, T_J(\omega)) - k(Y_i, T_J(\omega))) \in R^J, \quad (22)$$

$$W_n^\omega = \frac{1}{n} \sum_{i=1}^n Z_i^\omega \quad (23)$$

$$\Sigma_n^\omega = \frac{1}{n} \sum_i (Z_i^\omega - W_n^\omega)(Z_i^\omega - W_n^\omega)^T \quad (24)$$

$$S_n^\omega = n W_n^\omega \Sigma_n^{-1} W_n^\omega. \quad (25)$$

$$\quad (26)$$

If there exists $a \ne b$, such that $T_a(\omega) = T_b(\omega)$, then we redefine $Z_i^\omega = 0$.

Suppose $d_{\mu,J}^\omega(P, Q) = 0$. Then, by theorem 2, for all $j$, $\mu_P(T_j(\omega)) = \mu_Q(T_j(\omega))$. This implies that $\mathbb{E} Z_i^\omega = 0$, which in turn implies, by [2][5.2.3], that $S_n^\omega$ is asymptotically $\chi^2$ distributed with $J$ degrees of freedom.

If $\mathbb{E} Z_i^\omega \ne 0$ then

$$P(S_n^\omega > r) = P\left( (W_n^\omega)^\top (\Sigma_n^{-1})^\omega W_n^\omega - \frac{r}{n} > 0 \right) \to 1. \quad (27)$$

To see that, first we show that $(\Sigma_n^{-1})^\omega$ converges in probability to the positive definite matrix $(\Sigma^{-1})^\omega$. Indeed, each entry of the matrix $\Sigma_n^\omega$ converges to the matrix $\Sigma^\omega$, hence entires of the matrix $(\Sigma^{-1})^\omega$, given by a continuous function of the entries of $\Sigma^\omega$, are limit of the sequence $(\Sigma_n^{-1})^\omega$. Similarly $W_n^\omega$ converges in probability to the vector $W^\omega$. Since $(W^\omega)^\top (\Sigma^{-1})^\omega W^\omega = a^\omega > 0$ ($(\Sigma^{-1})^\omega$ is positive definite), then $(W_n^\omega)^\top (\Sigma_n^{-1})^\omega W_n^\omega$, being a continuous function of the entries of $W_n^\omega$ and $(\Sigma_n^{-1})^\omega$, converges to $a^\omega$. On the other hand $\frac{r}{n}$ converges to zero and the proposition follows. Finally since $d_{\mu,J}^\omega(P, Q) > 0$ almost surely then $\mathbb{E} Z_i^\omega \ne 0$ for almost all $\omega \in \Omega_1$.

We have showed that the proposition holds for almost all $\omega$, and thus $S_n^\omega$ converges for almost all $\omega$. Indeed it does not hold if it happens that for some $a \ne b$, $T_a(\omega) = T_b(\omega)$ or $d_{\mu,J}^\omega(P, Q) = 0$ for $P \ne Q$. But both those events have zero measure.

□

**Proof of Proposition 4** The poof is analogue to the proof of the Proposition 2.

## C   Other tests

### C.1   Quadratic-time MMD test

For two measures $P, Q$ the population $MMD$ can be written as

$$MMD(P,Q)^2 = \int k(x,x')dP(x)dP(x') - 2\int k(x,y)dP(x)dP(y) + \int k(y,y')dP(y)dP(y').$$

An MMD-based test uses as its statistic an empirical estimator of the squared population MMD, and rejects the null if this is larger than a threshold $r_\alpha$ corresponding to the $1-\alpha$ quantile of the null distribution. The minimum variance unbiased estimator of MMD is

$$MMD_n^2 = \frac{1}{\binom{n}{2}} \sum_{i \neq j} h(X_i, X_j, Y_i, Y_j),$$

$$h(x,x',y,y') = k(x,x') + k(y,y') - k(x,y') - k(x',y).$$

The test threshold $r_\alpha$ is costly to compute. The null distribution of $MMD_n^2$ is an infinite weighted sum of chi-squared random variables, where the weights are eigenvalues of the kernel with respect to the (unknown) distribution $P$. A bootstrap or permutation procedure may be used in obtaining consistent quantiles of the null distribution, however the cost is $O(b_n n^2)$ if we have $b_n$ permutations and $n$ data points ($b_n$ is usually in the hundreds, at minimum). As an alternative consistent procedure, the eigenvalues of the joint gram matrix over samples from $P$ and $Q$ may be used in place of the population eigenvalues; the fastest quadratic-time MMD test uses a gamma approximation to the null distribution, which works well most of the times, but has no consistency guarantees [10].

### C.2   Sub-quadratic time MMD test

An alternative to the quadratic-time MMD test is a B-test (block-based test): the idea is to break the data into blocks, compute a quadratic-time statistic on each block, and average these quantities to obtain the test statistic. More specifically, for an individual block, laying on the main diagonal and starting at position $(i-1)B+1$, the statistic $\eta(i)$ is calculated as

$$\eta(i) = \frac{1}{\binom{B}{2}} \sum_{a=(i-1)B+1}^{iB} \sum_{b=(i-1)B+1 \neq a}^{iB} h(X_a, X_b, Y_a, Y_b). \tag{28}$$

The overall test statistic is then

$$\eta = \frac{B}{n} \sum_{i=1}^{\frac{n}{B}} \eta(i). \tag{29}$$

The choice of $B$ determines computation time - at one extreme is the linear-time MMD suggested by [9, 11] where we have $n/2$ blocks of size $B=2$, and at the other extreme is the usual full MMD with 1 block of size $n$, which requires calculating the test statistic on the whole kernel matrix in quadratic time. In our case, the size of the block remains constant as $n$ increases, and is greater than 2. This is very similar to the case proposed by [32], and the consistency of the test is not affected.

B-test of [32] assumes that $B \to \infty$ together with $n$, which implies that the statistic $\hat\eta$ defined in (29) under the null distribution satisfies

$$\sqrt{nB}\hat\eta \xrightarrow{D} \mathcal{N}\left(0, 4\sigma_0^2\right), \tag{30}$$

for asymptotic variance $\sigma_0^2 = \mathbb{E}_{XX'}k^2(X,X') + (\mathbb{E}_{XX'}k(X,X'))^2 - 2\mathbb{E}_X\left[(\mathbb{E}_{X'}k(X,X'))^2\right]$ that can easily be estimated directly or by considering the empirical variance of the statistics computed within each of the blocks. Note that the same asymptotic variance $\sigma_0^2$ is obtained in the case of a quadratic-time statistic [9] – albeit convergence rate being a faster $O(1/n)$ in that case. Indeed, (30) is obtained directly from the leading term of the variance of each block-based statistic being $\frac{4\sigma_0^2}{B^2}$.

Figure D.3: Box plot of p-values used for parameter selection. The $X$ axis shows the binary logarithm of the scaling parameter applied to data. We chose the scaling with the smallest median p-value. If the medians were similar we used a scaling that had few outliers and was surrounded by other scalings with small p-values. In this example we chose scalings $\lambda = 2^0 = 1$ for the B-test, $\lambda = 2^{-8}$ for the Smoothed CF test, and $\lambda = 2^{-10}$ for the CF test.

Therefore, the p-value for B-test is approximated as $\Phi\left(-\frac{\sqrt{nB}\hat\eta}{2\hat\sigma_0}\right)$, where $\Phi$ is the standard normal cdf. When $B$ remains constant as $n$ increases, it can be shown that the variance of each block-based statistic is exactly $\frac{4\sigma_0^2}{B(B-1)}$, and thus we obtain by CLT that

$$\sqrt{n}\hat\eta \xrightarrow{D} \mathcal{N}\left(0, \frac{4\sigma_0^2}{B-1}\right).$$

Therefore, a slight change to p-value needs to be applied when $\sigma_0^2$ is estimated directly: $\Phi\left(-\frac{\sqrt{n(B-1)}\hat\eta}{2\hat\sigma_0}\right)$. If, however, one simply uses the empirical variance of the individual statistics computed within each block, the procedure is unaffected.

## D  Parameters Choice

We split our data into disjoint training and testing sets, and optimized parameters on the training set. To evaluate different data scalings $\lambda$, we plotted the associated p-values of tests on the training data. Figure $D$ presents such a plot for three different tests. The p-values were obtained by running the test several times (20 to 50) for each $\lambda$. In the case of simulatied data, we generated a new training dataset for each repetition at a given $\lambda$. For the amplitude modulated audio dataset, we added different independent noise to the training samples for each repetition (note that this was in retrospect not an ideal choice: a better approach would have been to draw bootstrap samples from the training data, possibly using additional tracks from the CD to provide sufficient training samples). For the Higgs dataset, we had an abundance of training data, hence we were able to use bootstrap samples without repetition from the training set. This last approach is the recommended strategy for real-life data. We emphasize that p-value optimization is a successful heuristic, but is *not* a substitute for a choice of parameters that optimizes test power. Better parameter choice might be accomplished following a strategy analogous to [11].