[Reviews · NeurIPS 2015]

Submitted by Assigned_Reviewer_1

== Summary ==

The paper presents nonparametric two-sample tests with a cost linear in the sample size. The proposed MMD formulation relies on the L2(R) distance between the characteristic functions of corresponding distributions where R is the density specified by the inverse Fourier transform of the kernel. This formulation was previously studied in Zhao & Meng (2014), for example. They showed that the original formulation fails to distinguish a large class of measures. To alleviate this problem, the paper presents a "smoothed" version of the formulation using an analytic smoothing kernel. The resulting metric is shown to be "random metric" which satisfies all the conditions for a metric with qualification "almost surely". The linear-time tests are obtained by approximating the integral with the sum over random samples from the inverse Fourier transform of the kernel. The new representation is a mapping of the space of probability measures into a finite dimensional Euclidean space that is almost surely an injection, and as a result almost surely a metrization.

Interestingly, the proposed linear-time test can outperform the quadratic-time MMD in terms of power of the test.

== Quality ==

The paper is technically sound. Although the general idea is not completely new, the proposed

method is supported by both theoretical guarantee and empirical results, which is a key strength of this paper.

comments/questions:

How does the proposed method depend on the kernel function? Intuitively, it seems there is an interaction between the behaviour of characteristic function and kernel. For example, what if the distribution is heavy-tailed? Is there any difference when using Gaussian or Laplace kernels?

== Clarity ==

The paper is clearly written. Minor comments below:

comments/questions:

Line 065: "... can be though as ..." => "... can be thought of as ..." ? Line 065: "... characteristics functions" => "... characteristic functions ..." Line 164: please check the last condition. Line 273: "... as ..." => "... are ..." ? Figure 1 is quite difficult to read as the colors are not distinguishable. Line 358: "... faster then ..." => "... faster than ..." Perhaps ref [32] can be updated to "D. M. Ji Zhao. FastMMD: Ensemble of circular discrepancy for efficient two-sample test. Neural Computation, 27(6):1345-1372, June 2015."

== Originality ==

The proposed idea is similar to that presented in

D. M. Ji Zhao. FastMMD: Ensemble of circular discrepancy for efficient two-sample test. Neural Computation, 27(6):1345-1372, June 2015.

The novelty of this work lies in the improvement of empirical characteristic formulation via "smoothing" which has been shown to improve the test statistic.

== Significance ==

Large-scale two-sample test has become increasingly important in both machine learning and statistics. This work provides an efficient tool for such a problem.

Summary: A neat idea for large-scale two sample-test with concrete contributions, both theoretically and empirically. Accept.

Submitted by Assigned_Reviewer_2

-weak review-

The authors propose a class of nonparametric two-sample independence tests of which they show theoretically and empirically that they are faster than baseline approaches.

The paper contains a lot of theory (didnt check the math) and several experiments, indicating increased statistcal power.

It looks like an accept (7)
Summary: Seems to be a solid paper (extensive math, neat idea, several first experiments).

Submitted by Assigned_Reviewer_3

This paper addresses an important problem of testing whether two distributions are different (i.e. two-sample tests which are ubiquitous in many data mining/machine learning applications).

It first introduces the maximum mean discrepancies method which is equivalent to the RKHS distance between mean embeddings of the two probability distributions. Naively computing the MMD between two distributions is quadratic in the sample size and a finite dimensional approximation based on random Fourier feature sampling is suggested. $J$ random Fourier features are sampled from the kernel and evaluated at the characteristic functions of the two distributions. This distance fails to distinguish a large number of probability distributions, however (Proposition 1), e.g. the injective property. Both of the newly proposed estimators turns the random Fourier feature finite dimensional approximation into a random metric. The difference is: given the random Fourier features (e.g. the frequencies to test the difference on), whether to evaluate the difference in characteristic functions or the to evaluate the difference in the mean embeddings.

The first proposed estimator evaluates the difference in characteristic functions. The key idea is to smooth the characteristic functions of the two distributions with an analytic kernel. Theorem 1 whose proof is in the appendix claims that this increases the distinguishing power of the modified distance measure. Then the authors demonstrate that the smoothed characteristic function can be expressed as an expected value of some function with respect to the given measure (a linear time estimator).

The second proposed estimator evaluates the difference in mean embeddings at the sampled random Fourier frequencies.

Hypothesis testing based on both of these estimators involve computation of test statistics which involves an inversion of $J$ by $J$ matrix (which the authors claim are small).

One minor note is that I found it confusing in the beginning to find how to compute the difference in smooth characteristic function in linear time, after introducing the metric based on mean embeddings. The authors should reorganize the paper regarding this point.
Summary: An interesting paper addressing fundamental problems in machine learning

Submitted by Assigned_Reviewer_4

Summary: This paper proposes two fast methods for nonparametric two sample testing. These are based on the notion of random metrics on distributions. The authors propose two approaches to make random metrics: one based on smooth characteristic functions, and one based of kernel mean embeddings with analytic kernels. Both the approaches essentially make use of representations of probabilities as analytic functions (in an RKHS). Analytic functions are different from each other almost surely w.r.t. any absolutely continuous distribution. Therefore almost surely unique representations of probabilities can be defined, by evaluating of the analytic representations of those probabilities on a random sample from an absolutely continuous distribution. By doing such evaluations on J random samples, the probabilities can be a.s. uniquely represented as J dimensional vectors. The random metrics are defined as distances between these J dimensional vectors. The resulting tests are fast to compute, and have high statistical powers.

I think this paper is nice and should be accepted. It is clever to represent distributions by finite dimensional vectors with analytic functions. I believe this approach can also speed up other methods based on kernel mean embeddings, and therefore would have significant impacts.

Quality: Both the theoretical analysis and experiments of the paper are well conducted. The following are some comments.

As J goes to infinity, Eq. (10) tends to || \mu_P - \mu_Q ||_{L2(\nu)}, where \nu is the absolutely continuous distribution of T_j. This further equals to || T^{1/2} (\mu_P - \mu_Q) ||_H, where T denotes the integral operator defined by the kernel and the distribution \nu. This can be seen as a smoothed version of MMD, as T^{1/2} has an effect of smoothing. Therefore Eq. (10) may be seen as an estimator of the smoothed MMD. This might explain why the test based on mean embedding performs well empirically for high-dimensional settings. Is this understanding correct? And if the space of the rebuttal permits, please explain how the performance change when J is increased.

Clarity: This is paper is well organized and easy to understand.

Originality: I believe that clever idea of using analytic functions to represent distributions is original.

Significance: The paper proposes approaches for representing distributions a.s. uniquely as finite dimensional vectors. I believe these are beneficial not only to two sample testing, but also to other problems of nonparametric statics. Examples include independence testing, Bayesian inference with graphical models, and reinforcement learning. These have been effectively tackled with kernel mean embeddings, but they are known to be slow. The approaches proposed in this paper can be used to speed up those kernel methods. Therefore I believe this paper would have significant impacts.

Minor points: Line 164: if P \neq Q then d(P,Q) > 0 a.s. ?
Summary: This paper proposes two fast methods for nonparametric two sample testing, based on almost surely unique representations of distributions as finite dimensional vectors based on analytic functions. I believe the proposed approach has significant impacts, as it is not only beneficial to two sample testing, but also to other methods based on kernel mean embeddings.

Author Feedback
Author rebuttal: ´╗┐We would like to thank the reviewers for the positive assessment of the paper and for the useful feedback provided. We apologize for the typos and instances of poor clarity which were raised - these will be corrected in the final version.

Reviewer 1
The relationship between kernel choice and test power will depend on the properties of the (unknown) distributions P and Q we are attempting to tell apart. One insight comes from looking at the differences between characteristic functions. The Laplace kernel has a Fourier transform which decays as 1/(1+t^2), and will give greater emphasis to differences in characteristic functions occurring at higher frequencies. The Gaussian kernel has a Fourier transform that decays exponentially, and focuses on differences at low frequencies. If such properties of P and Q are known a priori, the kernel may be chosen accordingly (of course, as [7] point out, if P and Q are fully known, then we will know precisely where their characteristic functions have the largest difference, but this is much stronger prior knowledge).

We will update the reference to D. M. Ji Zhao.

Reviewer 3
It is true that sometimes Smooth CF test outperforms the Mean Embeddings test, and indeed there is no reason in theory for either test to be uniformly more powerful. This is demonstrated in practice in our examples where for some cases, the smooth CF test is the more powerful test. The key benefit of our work, however, is that our new tests always give better power *per unit of computation* than other MMD-like quadratic or sub-quadratic tests. This performance/cost improvement is shown in experiments to be very substantial.

Reviewer 4
The interpretation of the equation 10 for large J is right, thank you for the new inside (| T^{1/2} (\mu_P - \mu_Q) ||_H). Indeed, it might be the reason for good performance in the high-dimensional settings, however since our J is small it is hard to argue that we approximate the integral || \mu_P - \mu_Q ||_{L2(\nu)} well.

Experiments suggest that increasing J increases the power but eventually decreases the power/time ratio.

Reviewer 5
The matrices that we invert are at most 10x10, since we use J at most 10. As outlined above, larger J don't increase the power much but increase the computational time substantially.

We will reorganize paper so that it will be clear from the beginning that difference in smooth characteristic functions can be calculated in linear time.